# Long-Term Evaluation and Calibration of Low-Cost Particulate Matter (PM) Sensor

**DOI:** 10.3390/s20133617

**Published:** 2020-06-27

**Authors:** Hoochang Lee, Jiseock Kang, Sungjung Kim, Yunseok Im, Seungsung Yoo, Dongjun Lee

**Affiliations:** 1Department of Mechanical Engineering, Seoul National University, 1 Gwanak-ro, Gwanak-gu, Seoul 08826, Korea; chang.lee@snu.ac.kr (H.L.); jskang0894@snu.ac.kr (J.K.); sjkim90@snu.ac.kr (S.K.); 2Air Quality Analysis and Control Center, Seoul Metropolitan Research Institute of Public Health and Environment, 30, Janggunmaeul 3-gil, Gwacheon-si, Gyeonggi-do, Seoul 08826, Korea; imyunseok@seoul.go.kr (Y.I.); yooss323@seoul.go.kr (S.Y.)

**Keywords:** particulate matter (PM), low-cost sensor, calibration, multivariate linear regression (MLR), multilayer perceptron (MLP), segmented model and residual treatment (SMART) calibration

## Abstract

Low-cost light scattering particulate matter (PM) sensors have been widely researched and deployed in order to overcome the limitations of low spatio-temporal resolution of government-operated beta attenuation monitor (BAM). However, the accuracy of low-cost sensors has been questioned, thus impeding their wide adoption in practice. To evaluate the accuracy of low-cost PM sensors in the field, a multi-sensor platform has been developed and co-located with BAM in Dongjak-gu, Seoul, Korea from 15 January 2019 to 4 September 2019. In this paper, a sample variation of low-cost sensors has been analyzed while using three commercial low-cost PM sensors. Influences on PM sensor by environmental conditions, such as humidity, temperature, and ambient light, have also been described. Based on this information, we developed a novel combined calibration algorithm, which selectively applies multiple calibration models and statistically reduces residuals, while using a prebuilt parameter lookup table where each cell records statistical parameters of each calibration model at current input parameters. As our proposed framework significantly improves the accuracy of the low-cost PM sensors (e.g., RMSE: 23.94 → 4.70 μg/m${}^{3}$) and increases the correlation (e.g., R${}^{2}$: 0.41 → 0.89), this calibration model can be transferred to all sensor nodes through the sensor network.

## 1. Introduction

Particulate matter (PM) is classified by size bins of maximum aerodynamic diameter (e.g., PM10 < 10 μm, PM2.5 < 2.5 μm, and PM1.0 < 1 μm). Exposure to PM is regarded as a major health risk and it causes various diseases from respiratory and cardiovascular diseases to neurodevelopmental disorders and mental disorders [1]. According to recent reviews, it globally affects a mortality rate of up to 4.2 million deaths per year [2,3]. The collection and analysis of PM concentration data is now being major interest of government and non-government organizations because of such an effect on public health. Meanwhile PM concentration features spatial and temporal fluctuation due to their aerodynamic nature, hence enabling higher spatiotemporal resolution of the PM concentration data is also being increasingly important. However, maintaining such high resolution with a government-grade air monitoring station is nearly impossible by the matter of cost. Additionally, their sampling interval is rather long, at the cost of the data quality. Because of the above facts, low-cost light scattering PM sensor have been widely used for a practical alternative of the air monitoring station in dense sensor deployment [4]. Even though these sensors still have a major challenge on data quality, they have overwhelming advantages of less expensive price, more compact size, and faster update rate [5,6]. As a result, many countries have densely deployed the low-cost sensor in the smart city [7,8,9]. As of April 2020, there are 40 government-operating beta attenuation monitor (BAM) stations in Seoul releasing information to the public every hour [10]. Additionally, approximately 3500 light-scattering PM equipment have been deployed in Korean major cities by leading telecommunication companies [11,12] and have continuously increased spatiotemporal resolution, as shown in Figure 1. As the importance of low-cost sensors has been increasing, more research is being conducted to evaluate and calibrate low-cost light scattering sensors.

Evaluation of low-cost sensors was analyzed under various climate and weather conditions over the world from a day to longer than a year [13,14,15]. Additionally, these studies have several aims, such as environmental effect analysis [16], newly developed sensor validation [17], and calibration performance evaluation. We built four kinds of rough prototypes for briefly checking sample-to-sample variability (PMSA003, PMS7003 (Plantower Inc., Beijing, China [18]), SEN0177 (DFRobot Inc., Shanghai, China [19]), and HPMA115s0 (Honeywell Sensing Inc., Charlotte, NC, USA [20])). Subsequently, we chose PMS7003 and developed a muti-sensor platform for further long-term evaluation. We describe performance limitation on the raw signal of low-cost sensors that have been identified by co-locating them with governmental BAM for about 7.5 months in Section 3.2. Plus, we compared theperformance between raw signal and calibrated signals under various environmental explanatory variables, sampling intervals, and calibration methods.

Based on the previous research of low-cost PM sensors [13,14,15,16,17], the low-cost sensor has limited accuracy and it requires a calibration procedure in order to boost accuracy. The most common calibration methods on PM2.5 calibration are a linear calibration accounting for two-thirds of total calibration cases according to a technical report from the Joint Research Center of the European Commission [21] (univariate linear regression (ULR)—46%/multivariate linear regression (MLR)—22%). As such, linear regression (LR) is widely used for PM calibration, since it is a simple and powerful method. However, LR sometimes generates an under-fitting problem when the true function of data is not sufficient to fit the linear function approximation. For example, MLR suffers severe performance degradation under a high humidity environment [22]. On the other hand, non-linear calibration is quite free from the problem, but it is required to avoid an over-fitting problem by selecting an appropriate order of function approximation.

Beyond the cases of a single calibration model, sequentially combined calibration models were studied. Lin et al. 2018 introduced a two-phase calibration model while using Akaike information criterion (AIC) and random forests (RF). As a first phase, several linear models are created by selecting subsets from the entire input variable space based on the AIC index. After that, RF is used to learn the residual of the linear models [23]. However, RF uses the aggregation of randomized models with several decision trees and their results are averaged in the regression problem; it is usually good at avoiding over-fitting problems, but it might present lower accuracy due to the averaged result from several decision trees. Cordero et al. 2018 obtained the calibrated PM value through the linear model to generate the difference of the raw PM value. Subsequently, a non-linear calibration among RF, support vector machine (SVM), and artificial neural networks (ANN) is performed using the difference and the input variables [24]. However, their dataset was small and the training dataset and test dataset were shared with the k-fold cross-validation method.

This paper introduces a novel combined calibration method that selects the most accurate model from models for each sampling. This combined calibration differs from the cited methods in dividing the entire input variable space into segmented cells and applying the best model among multiple models for each cell. Besides, we proposed additional procedures to reduce the residuals probabilistically by managing the sum of residuals that are generated by the selected model in each cell. This combined calibration is named segmented model and residual treatment calibration (SMART calibration). The performance of this SMART calibration method was analyzed with raw data and compared with not only other state-of-the-art calibration methods, but also other study group’s calibrated results based on 16 month-duration datasets [25]. The comparison results show that our proposed method offers better accuracy than counterparts.

Our contribution can be summarized, as follows:Field evaluation of low-cost PM2.5 sensor in Seoul, Korea has been executed and analyzed. These were under several conditions, such as environmental explanatory variables (humidity/temperature/ambient light), sampling intervals (5 min/1 h/24 h), and calibration methods (linear/non-linear/SMART calibration).A novel combined calibration method has been introduced to increase low-cost sensor accuracy. The performance was compared to other calibration methods. This calibration method can also be applied to an upcoming future dataset with the previously generated models.

The next sections are structured, as follows. Section 2 describes the overall method of this research including data collection, data preprocessing, and data calibration. Section 3 presents the results and discussion. It covers the result of the experiments and explains the analysis of the result. Section 4 summarizes this paper and explains the potential use cases.

## 2. Methods

This section is written for describing the overall procedures of evaluation and calibration on low-cost sensors. It includes data collection (Section 2.1), data preprocessing (Section 2.2), data calibration methods (Section 2.3), and metric information (Section 2.4). Figure 2 shows the overall procedures for sensor evaluation and calibration. A multi-sensor platform has been developed and co-located with the governmental BAM in the government station (Dongjak-gu, Seoul, Korea) to evaluate low-cost light scattering PM2.5 sensor. The data have been collected for around 7.5 months (15 January 2019–4 September 2019). The following subsections will explain more information on several procedures we executed.

### 2.1. Data Collection

In this section, the sensor configuration and deployment information on the low-cost sensor and reference system is described.

#### 2.1.1. Multi-Sensor Platform—Low-Cost Light Scattering PM Sensor

We developed prototypes and roughly evaluated the repeatability of signal and the sample-to-sample variability to select a proper low-cost sensor among four kinds of commercial low-cost sensors. Based on this analysis, PMS7003 (Plantower Inc., Beijing, China [18]) was chosen and the configuration and design of the multi-sensor system development proceeded for long-term evaluation and calibration.Detailed information of prototypes evaluation is further described in Appendix C. The selected PM sensor and other environmental sensors were built together as a multi-sensor platform, as shown in Figure 3a.

Three low-cost PM sensors are mounted on a single multi-sensor platform to identify sample variation among three low-cost sensor samples. It also includes environmental sensors of humidity, temperature, and ambient light to analyze and calibrate the environmental impact on the measurement of PM. Data collection of each sensor module in low level is performed through Arduino Due, and communication with sensor network in high level is implemented through Raspberry Pi 3B+, as shown in Figure 3b. Data are measured and stored at 1-s sampling intervals and configured to be transmitted to users via wired LAN or Wi-Fi.

#### 2.1.2. Governmental BAM—High-End PM Monitoring Station

In Korea, BAM is the only regulatory reference that received a formal approval from the Korean Ministry of Environment. As a reference to the experiment, the PM711 model (Kimoto Inc., Osaka, Japan [26]) was selected because it has a relatively fast sampling interval (5 min) compared to a sampling interval (1 h) of other BAM as shown in Figure 4b. Five min. sampled output may be less accurate than 1 h averaged output since 5 min. sampling interval data is the data source of 1 h averaged output. This equipment consists of two separate racks of monitoring systems for PM2.5 and PM10 measurements. It features a high accuracy, since it includes a sampling stabilizer, such as particle separator (PM2.5 impactor and PM10 impactor) and environment controllers of temperature, humidity, and air-flow to stably supply PM.

### 2.2. Data Preprocessing

In this step, different data sampling intervals of two equipment were matched so that the data from the multi-sensor platform can be directly compared with the data from the governmental BAM. Data were excluded from data preprocessing if any intermittent data were observed from sensor modules. The data of the multi-sensor platform was averaged with a 5-min. fixed window. The preconditioned data were used to build the linear/non-linear calibration model, such as MLR, MLP, and SMART calibration, and to perform the actual calibration with the prebuilt model in the next step. To build and evaluate the calibration model, the dataset was constructed in two ways for comparison, as shown in Figure 5. One is sampling in a sequential manner (hereinafter sequential) and the other is a random manner (hereinafter shuffled) under various separating ratio (unless otherwise stated, 80% of the total datasets were randomly selected to construct a training dataset and the remaining 20% was used as a test dataset). Data preprocessing was done via Matlab R2018b [27] and Python 3. Pandas [28], the state-of-the-art Python data manipulation library, was also utilized for data preprocessing.

### 2.3. Data Calibration

In this paper, calibration doesn’t mean any correction for the observed data in the training dataset. The calibration means an estimation for the unseen data in the training dataset. PM2.5 (low-cost sensor), humidity, temperature, and ambient light were selected as explanatory variables, and PM2.5 (BAM) was selected as a response variable. The influence of each explanatory variable was separately analyzed in Section 3.3. The calibration methods were analyzed in three ways: linear, non-linear, and SMART calibration. Data calibration was performed via Python 3 libraries (pandas [28], keras [29], sklearn [30] and tensorflow [31]).

#### 2.3.1. Linear Calibration

Based on multivariate linear regression (MLR), we selected PM (low-cost sensor), humidity, and temperature of the multi-sensor platform as explanatory variables and chose PM (governmental BAM) as the response variable. The least-square method was applied with the chosen coefficients as shown in Table 1 (all the p-values for each coefficient were all less than 0.00001 and are omitted hereinafter.)
(1)$$\begin{array}{c} \hat{y}=w_{0}+\sum_{i=1}^{N}w_{i}x_{i}\\ \hat{y}:PM\underline{\;}calibrated,\;w_{0}:intercept,\;w_{i}:coefficient,\;x_{i}:input\;variable\underline{\;}measured \end{array}$$


#### 2.3.2. Nonlinear Calibration

Non-linear calibration was performed based on a multilayer perceptron (MLP) from the neural network and it consists of an input layer, an output layer, and hidden layers. The calibration is performed by making an appropriate sum of weights between neurons existing in each layer, as shown in Figure 6. The sum of each weight passes a non-linear activation function, rectified linear unit (ReLU), to generate a non-linear model. ReLU activation is explained in Equation (Equation 2). PM2.5 (low-cost sensor), humidity, and temperature from the multi-sensor platform were preprocessed and used as input variables in the input layer. PM2.5 (BAM) from the governmental station was used as output variables in the output layer. Hyperparameters were manually chosen under several trials, as shown in Table 2.
(2)$$\begin{array}{c} \hat{y}=W_{3}max\left(0,\;W_{2}max\left(0,\;W_{1}x\right)\right)\\ \hat{y}:PM\underline{\;}calibrated,\;W_{i}:weight\;matrix,\;x:input\;variable\underline{\;}measured \end{array}$$


#### 2.3.3. SMART Calibration (Combined Calibration)

In this section, we introduce a SMART calibration algorithm, which selectively maps most probabilistically appropriate models given multiple linear/non-linear calibration models. LR is the most representative methodology for finding a best-fit line for the approximation and estimation. However, the LR is usually too simple to correctly fit the true function of complex data. And the best-fit line is highly affected by non-linearity, outliers, and data range. Meanwhile, non-linear calibration can optimally generate a model which has lower prediction error of training dataset as the model complexity increases more. However, in this case, a prediction error of the test dataset is largely generated in case the model is overfitted. This is well known disadvantage of non-linear calibration (limitations of linear and non-linear calibration are further described in Appendix A).

Each model has its “weak spot” in their domain due to the above nature of the linear/non-linear calibration models. For instance, LR has its weak spot in the non-linear region of the domain, and MLP has weak spot in the overfitted region. The SMART calibration method has been developed to improve this limitation. Figure 7 shows the overall procedures of model build and model selection. Firstly, two training models and residual maps are generated with training dataset in model build step. Secondly, a prevailing model map is constructed by comparing residual maps. Subsequently, the prevailing model map can be utilized in the model selection step.

In more detail, the residual map that divides a full range of explanatory variable space (e.g., temperature and humidity) into segmented small area cells is generated, as shown in Figure 8. Every residual of training data is allocated to a corresponding partitioned cell of residual maps. The distribution of residuals in each cell of a residual map are assumed as a Gaussian, since residual is the error of the estimator. Each cell has its probability density function (PDF), which is expressed by its average and standard deviation. This information is stored in residual maps. For each cell, a prevailing calibration model is defined by comparing the residual maps of the linear and non-linear models. Every prevailing calibration model of each cell is stored in a prevailing model map. Once a prevailing model map is completed through a whole training dataset, the corresponding input cell of test dataset calibrates their data with a predefined suitable model and averaged residual, as shown in Figure 9 (Procedures of SMART calibration are further described in Appendix D). Figure 7, Figure 8 and Figure 9 are examples of explanations and the number and type of calibration models are not limited in MLR and MLP. SMART calibration has good features on the simpleness of procedures and the compatibility of several models since it is the hierarchical calibration model. As it depends on the consistency of estimators, the number of data in each cell is increased when the accuracy of SMART calibration is increased. Additionally, it has good performance with a high bias model, but it cannot outperform when SMART has only high variance models, since SMART calibration selects model according to variance of data in segmented cell.

### 2.4. Metric Information

Four key metrics were used to analyze the performance as shown in Table 3. The analysis index used mean absolute error (MAE), mean squared error (MSE), root mean squared error (RMSE), and R${}^{2}$ (coefficient of determinant). RMSE is excluded hereinafter, because it can be calculated by MSE. In some analysis cases, slope, intercept, mean and standard deviation, quartile, and Pearson’s correlation coefficient are also used.

## 3. Results and Discussions

This section is written for describing preliminary analysis (Section 3.1) by varying explanatory variables and sampling interval conditions. Subsequently, we compare the performance of SMART calibration under several conditions, such as before calibration (Section 3.2), after calibration (Section 3.3), other calibrations methods (Section 3.4), and a previous similar study (Section 3.5).

### 3.1. Preliminary Analysis

#### 3.1.1. Performance Characteristics: Explanatory Variables

The low-cost sensor features cost-effectiveness, lightweight, rapid, and continuous measurements, but it has a limitation on their accuracy. Accordingly, this low-cost sensor generally excludes any sampling stabilizer for PM size, humidity, temperature, or flow control. As a result, the low-cost sensor is directly affected by the surrounding environment. In particular, the influence of humidity and temperature has been continuously researched by several research groups, and calibration models that are based on meteorological parameters are introduced as Equations (Equation 3) and (Equation 4) [22,32].
(3)$$\hat{y}=\beta_{1}+\beta_{2}\frac{\rho^{2}}{\left(1-\rho \right)}y+\beta_{0}$$
(4)$$\hat{y}=\alpha_{1}y+\alpha_{2}t+\alpha_{0}$$

$\hat{y}:PM\underline{\;}calibrated,\;\;\alpha_{i}:coefficient,\;\;y:PM\underline{\;}measured,\;\;\rho :RH\underline{\;}measured,\;\;t=temp.\underline{\;}measured$.

In this section, short-term analysis for the effects of humidity, temperature, and ambient light on PM concentration was performed, and long-term analysis for the effects of humidity and temperature was executed while applying linear and non-linear calibration. As a result, we found that the humidity and temperature is the important variable on PM concentration calibration.

##### Performance Characteristics: Explanatory Variables, Short-Term Analysis (45 Days)

The experimental data from 18 July 2019 to 4 September 2019 were analyzed, since the storage of data on the ambient light sensor was executed in this limited period. This period was summer in Korea and the summer climate of Korea is characterized by high temperatures and high humidity. As previously researched in Equation (Equation 3), high humidity features high non-linearity of the calibration function. In our result, the non-linear calibration had a relatively smaller error than the linear calibration, as shown in Table 4.

The comparison of the uncalibrated raw PM signal and the calibrated PM signal expressed a significant improvement (e.g., MAE of MLP: 9.78 → 3.55 μg/m${}^{3}$), and the calibration, including the PM raw signal with humidity signal showed remarkable improvement (e.g., MAE of MLP: 3.55 → 2.99 μg/m${}^{3}$). In the case of calibrations, including temperature and ambient light, the improvement was insignificant. The long-term analysis was performed in the next section on the influence of PM, humidity, and temperature.

##### Performance Characteristics: Explanatory Variables, Long-Term Analysis (7.5 Months)

The experimental data from 15 January 2019 to 4 September 2019 were analyzed in Table 5. Similar to short-term analysis, the uncalibrated raw PM signal and the calibrated PM signal (e.g., MAE: 15.87 → 4.21 μg/m${}^{3}$), and the calibration, including raw PM signal with humidity signal (e.g., MAE: 4.21 → 4.04 μg/m${}^{3}$) showed a significant improvement. The performance by humidity signal under the short-term analysis was highly improved where the high humidity region accounted for the majority, whereas, under the long-term, analysis was slightly improved. However, the performance was highly improved by adding temperature, especially for non-linear calibration cases (e.g., MAE: 4.04 → 3.52 μg/m${}^{3}$).

#### 3.1.2. Performance Characteristics: Sampling Interval

In this section, the 5 min. sampling interval was converted into one hour and 24 h sampling interval to compare with other previous studies. Most of the PM researchers analyzed sensor performance under one hour or 24 h of sampling interval, because the high-end BAM as a reference-grade instrument was used in hourly sampling intervals. Especially, Met One BAM-1020 (Met One instrument Inc., Grants Pass, OR, USA [33]), a US EPA [34] certified equipment, was used in many previous studies [14,25].

Non-overlapping sliding windows were applied for one hour or 24 h of sampling intervals. MAE decreased with longer sampling intervals, since more aggregated data reduced data variation, as shown in Table 6. In the case of the 24 h sampling interval, R${}^{2}$, which indicates proportional variance for response variables was lowered. This lowered R${}^{2}$ is derived from reduced data range by aggregation. This can be calculated from the R${}^{2}$ equation in Table 3 or explained by Figure 10.

### 3.2. Comparative Analysis: The Low-Sensor and Governmental BAM (Before Calibration)

The performance of the low-cost sensor was analyzed by comparing raw signals from the sensor platform and the reference signal from the governmental BAM (hereinafter three low-cost sensors’ raw signals are described as Raw (a/b/c), and the BAM signal is remarked as BAM in Tables and Figures). Figure 11 shows the correlation between three low-cost sensors and the BAM. Additionally, their correlation coefficient, evaluation metrics, and statistic summary are listed in Appendix B (Table A1 and Table A2).

R${}^{2}$ of low-cost sensors with BAM was expressed as 0.416, 0.546, and 0.417. However, R${}^{2}$ among low-cost sensors expressed a very strong positive correlation coefficient, with 0.937, 0.994, and 0.933. It is possible to expect the effectiveness of the performance improvement via the calibration due to a very strong correlation coefficient with BAM output. Additionally, high R${}^{2}$ among the low-cost sensors in the commonplace indicates that a common calibration model can be shared under logged condition. The data distribution expresses the overall difference between the low-cost sensor and the BAM, as shown in Figure 11. The reproducibility among the low-cost sensors looked high with a very tight output span, but the reproducibility between the low-cost sensors with the BAM output looked low with a wide output span.

### 3.3. Comparative Analysis: The Low-Cost Sensor and Governmental BAM (After Calibration)

MLR, MLP, and SMART calibration were executed to evaluate the performance by following the methods in Section 2 with a PM sensor instead of three PM sensors. All of the described results from this subsection were only calculated by the test dataset, since the training dataset was used for calibration model generation. Figure 12 shows the correlation between low-cost sensors and BAM. Additionally, their correlation coefficient, evaluation metrics, and statistic summary are listed in Appendix B (Table A3 and Table A4).

The means and standard deviations in 38.12±31.18
μg/m${}^{3}$ (raw signal), 23.13±13.74
μg/m${}^{3}$ (MLR), 22.7±13.12
μg/m${}^{3}$ (MLP), and 23.09±13.85
μg/m${}^{3}$ (SMART calibration) were obtained and compared with 23.10±14.84
μg/m${}^{3}$ (BAM). The normalized mean bias error declined from 65% to 1.7% and standard deviation decreased from 110% to 11.6% by applying MLP calibration models. R${}^{2}$ were observed as 0.41 (raw signal), 0.84 (MLR), 0.86 (MLP), and 0.89 (SMART calibration), respectively. By these results, the calibration significantly improves the performance of the low-cost sensors.

As shown in Figure 13 and Table 7, several calibration results were analyzed by applying different data preprocessing conditions. Our dataset was analyzed by a shuffled method as well as a sequential method, since Korea has four distinct seasons and 7.5 months collected dataset was experienced through the limited climate and season. The shuffled dataset features a higher R${}^{2}$ than the sequential dataset. On the other hand, the sequential dataset features a lower error in MAE and MSE than the calibration result of the dataset under the shuffled condition. Appendix E further describes more information on several shuffled methods on successive hourly or daily data chunk size.

This calibration can also be applied to an upcoming future dataset with the previously generated calibration models under the sequential method. As an example, the sequential datasets from the raw signal, the SMART calibration signal, and the government BAM’s signal were plotted, as shown in Figure 14. For detailed information, a training dataset was constructed with the sequential condition from 15 January 2019 to 8 August 2019 and their calibration model was created. After that, the test dataset was built from 8 August 2019 to 4 September 2019 and the previously derived model from the training dataset was applied. As a result, the test dataset confirms a very similar BAM output (e.g., MAE = 2.79, MSE = 14.02, and R${}^{2}$ = 0.76).

### 3.4. Comparative Analysis: Other Calibration Methods

The SMART calibration method was compared with other regression methods, such as lasso regularization, ridge regularization, and polynomial linear regression (PLR). Additionally, we applied state-of-the-art ensemble learning methods such as random forests (RF), extreme gradient boosting (XGB), and light gradient boosting (LGB). The hyperparameters of these methods were exhaustively searched over specified hyperparameters. A cross-validated grid search algorithm was applied in order to optimize hyperparameter and more information of the hyperparameter grid is further described in Appendix F. SMART calibration parameters were also customized with an increased cell size of the residual map and another calibration model. Several dataset ratios under the sequential method were applied for the data precondition method. Our calibration method expressed the smallest MAE and MSE among twelve calibration methods, as shown in Figure 15 and Table 8.

### 3.5. Comparative Analysis: Previous Similar Study

The SMART calibration result was compared with the latest results from a similar study because we could not get a long-term dataset of other research under similar conditions [25]. The study had a field test for 16 months in North Carolina, USA by comparing a commercial product (PA-II (Purple Air Inc., Draper, UT, USA [36])) with a BAM 1020 (Met One instrument Inc., Grants Pass, OR, USA [33]). This study included a long-term performance evaluation and a calibration under 1 h sampling interval basis. 90% training dataset and 10% test dataset by the shuffled (random) method was conducted in data preprocessing. MLR with raw PM signal, humidity, and temperature was applied for their calibration method.

Before the calibration, the results of the other group study were superior, thanks to a factory calibration under product manufacturing, as shown in Table 9. After the calibration, our group’s shuffled dataset showed higher R${}^{2}$ than the other group study and our group’s sequential dataset with SMART calibration was superior in all performance aspects.

## 4. Conclusions

The low-cost PM sensor was evaluated and it was calibrated with co-located governmental BAM in the urban air monitoring station (Dongjak-gu, Seoul, Korea). The performance of the low-cost PM sensor was analyzed using the analysis metrics of MAE, MSE, RMSE, R${}^{2}$, slope, intercept, mean, standard deviation, and quartile. The means and standard deviations in the raw signal of the low-cost sensor and BAM output were 38.15±31.29 and 23.10±14.84
μg/m${}^{3}$, with around 65% normalized mean bias error. Additionally, a comparison of calibration methods, such as MLR, MLP, and SMART calibration, was performed. The means and standard deviations in the SMART calibration of the low-cost sensor and the BAM output were 23.09±13.85 and 23.01±14.74
μg/m${}^{3}$ with around 0.35% normalized mean bias error. When the raw signal and calibrated signals of the low-cost sensor were compared to the figures from BAM output by applying correlation index, R${}^{2}$, increased correlations between the low-cost sensor and the BAM output were observed as 0.41 (raw signal), 0.82 (LR), 0.84 (MLR), 0.83 (MLP), and 0.89 (SMART calibration). Furthermore, this calibration model was verified with the possibility of being applied to future datasets. These results explain the fact that calibration is highly required when low-cost sensors are used for high accuracy sensing.

A sample-to-sample variability of the low-cost sensors was evaluated among three co-located low-cost sensors. The sensors were very strongly correlated having an extremely high correlation coefficient ranging from 0.985 to 0.997. Based on this finding, a calibration model can be continuously updated and improved by co-locating a single multi-sensor platform with BAM and it can be transferred toward all nodes in a sensor network to calibrate the entire nodes. This approach is the base concept of an online calibration for low-cost sensors. For future studies, a mobile node that is converted from the co-located multi-sensor platform travels among all of the nodes in the sensor network by performing an offline calibration of slope and intercept of each node. This successive calibration is named Hybrid Calibration, which features both an entire online calibration and an individual offline calibration. 

## Figures and Tables

**Figure 1 sensors-20-03617-f001:**
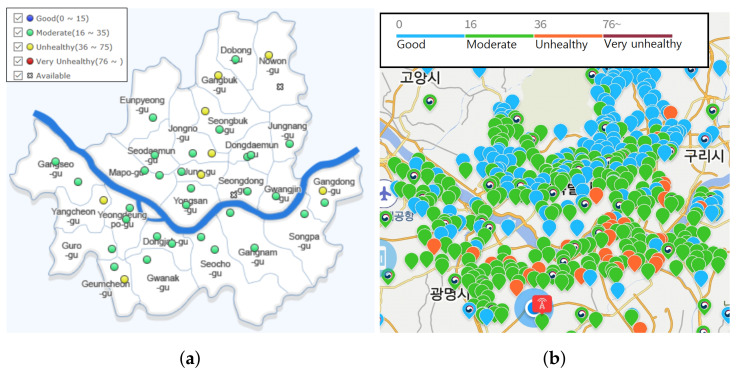
Comparison of deployment density by the responsible organization in Seoul. A circle indicates the location of equipment with Korea air quality index (AQI) of PM2.5. (**a**) By government (BAM); (**b**) By a company (Light-scattering) [12].

**Figure 2 sensors-20-03617-f002:**
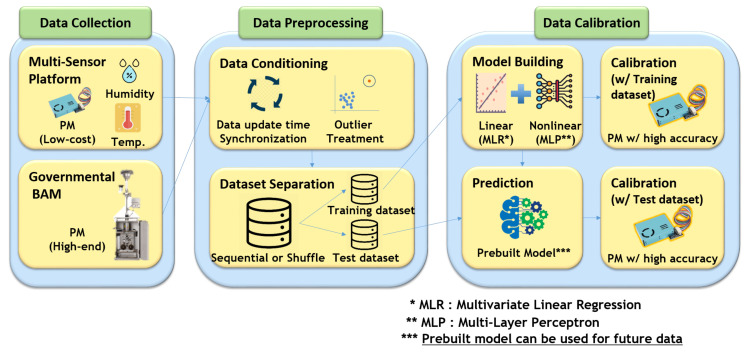
Overall procedures for sensor evaluation and calibration.

**Figure 3 sensors-20-03617-f003:**
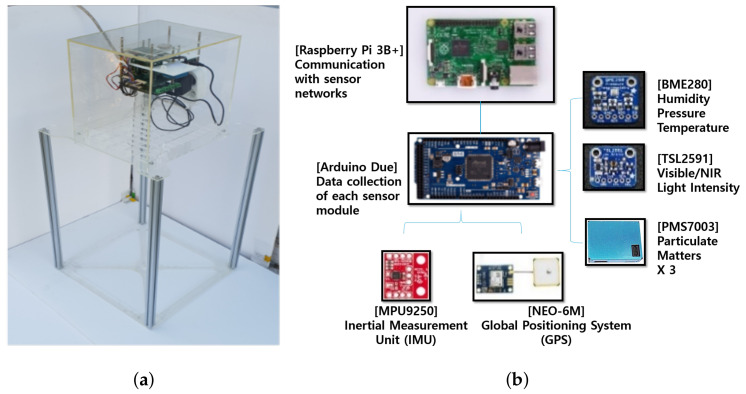
Information on multi-sensor platform. (**a**) Picture of platform; (**b**) Configuration of submodules.

**Figure 4 sensors-20-03617-f004:**
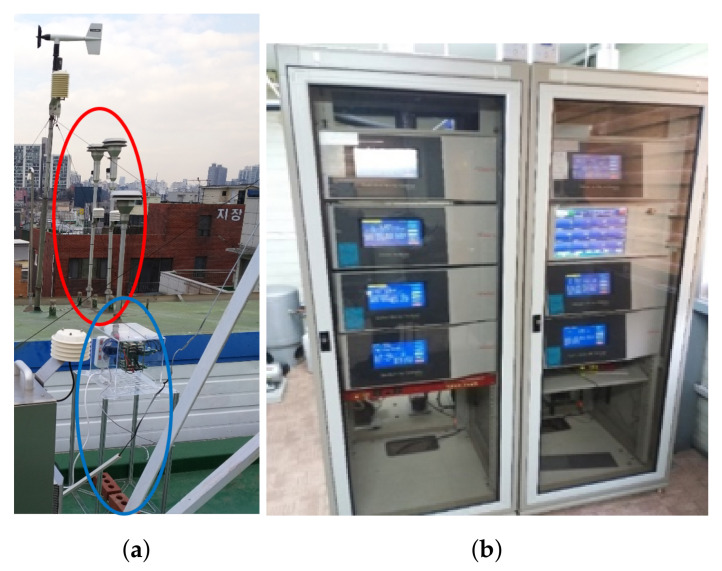
Information on governmental BAM station. (**a**) Outside; (**b**) Inside. It’s located at 6, Sadang-ro 16a-gil, Dongjak-gu, Seoul, Korea, and operated by the Seoul research institute of public health and environment. Inlets of BAM (red circle) and Multi-sensor platform (blue circle) are located together.

**Figure 5 sensors-20-03617-f005:**
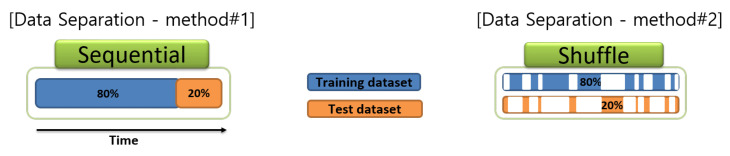
Default data separation methods for the training dataset and test dataset. 20% of the training dataset is used for the validation dataset to prevent the over-fitting calibration model. A shuffled method is controlled by a fixed random seed to compare the performance between calibration algorithms.

**Figure 6 sensors-20-03617-f006:**
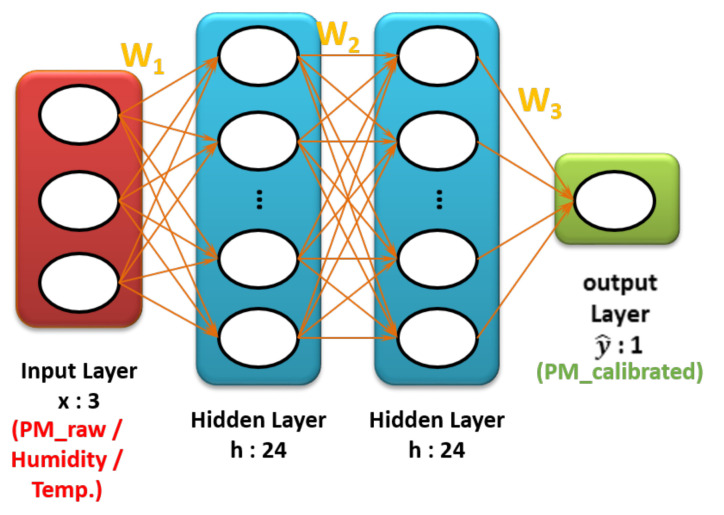
The architecture of a fully connected neural network. An input layer in red feeds explanatory variables and an output layer in green feeds response variable. Based on hyperparameter, the weight matrix (parameter) is built.

**Figure 7 sensors-20-03617-f007:**
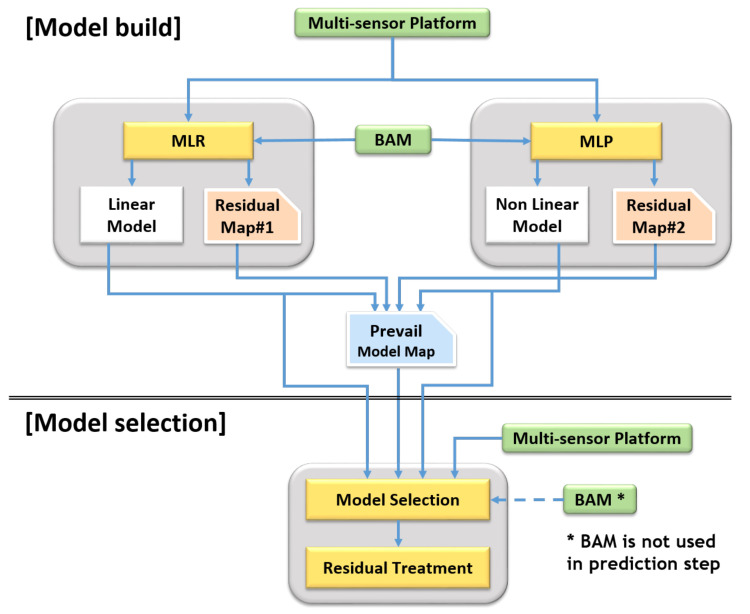
Overall procedures for SMART calibration. (e.g., MLR and multilayer perceptron (MLP) model).

**Figure 8 sensors-20-03617-f008:**
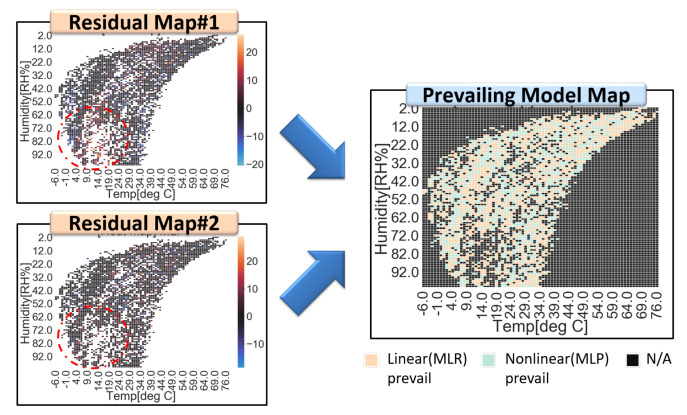
Residual maps and a prevailing model map for SMART calibration. Residual map#1 from the linear model (**top left**) and residual map#2 from the nonlinear model (**bottom left**) are merged into a prevailing model map (**right**). Residuals under high humidity and low-temperature condition are indicated in red dotted circles. In this region, the linear model has higher expectations of residuals than the nonlinear model.

**Figure 9 sensors-20-03617-f009:**
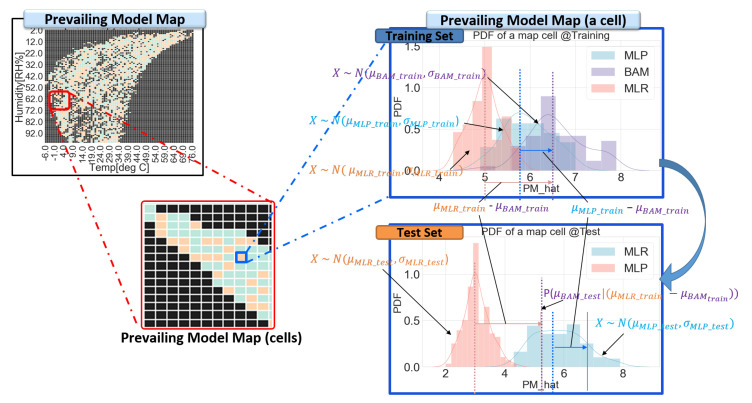
Prevailing model map and prevailing model map (segmented cell). A cell (**Blue box**) is segmented by the allocated inputs, and it has means and standard deviations of calibration models and BAM. The cell offers the prevailing model and its residual for the allocated input.

**Figure 10 sensors-20-03617-f010:**
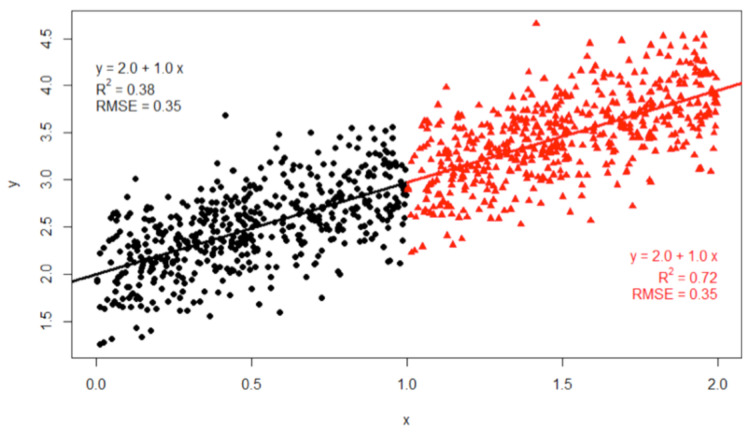
A characteristic of R${}^{2}$ according to widen data range. The black dot alone has R${}^{2}$ = 0.38, while the black dot and red triangle have R${}^{2}$ = 0.72. However, their regression line and RMSE are the same (Figure quoted [35]). This descriptive statistic is also required when R${}^{2}$ is used as a metric.

**Figure 11 sensors-20-03617-f011:**
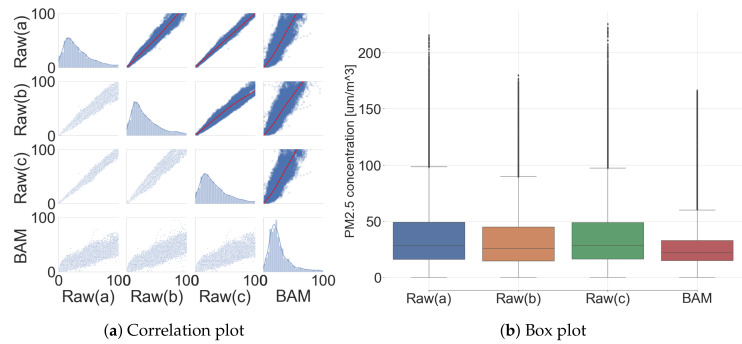
Comparison between low-cost sensors and governmental BAM (before calibration).

**Figure 12 sensors-20-03617-f012:**
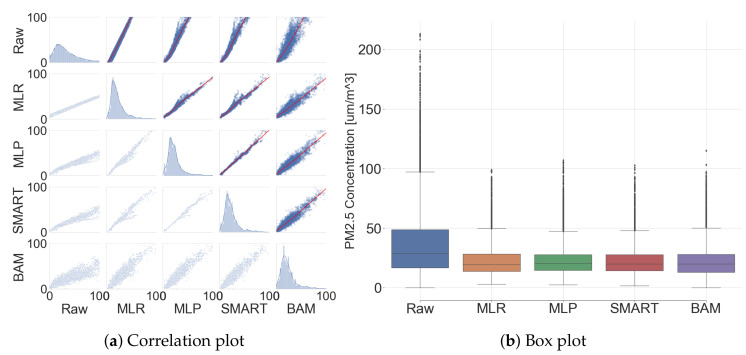
Comparison between low-cost sensors and governmental BAM (after calibration).

**Figure 13 sensors-20-03617-f013:**
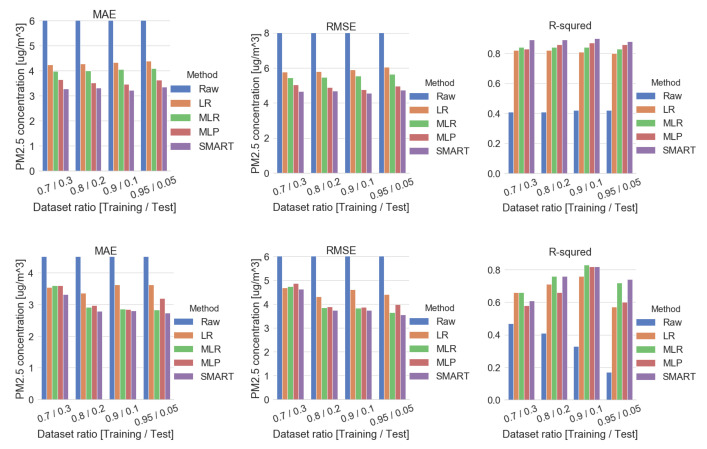
Comparison plot by data preprocessing methods—shuffled (**top**) and sequential (**bottom**).

**Figure 14 sensors-20-03617-f014:**
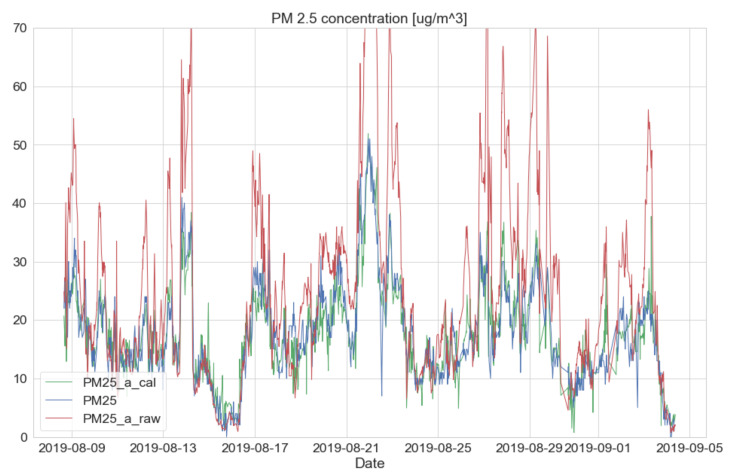
Comparison of output plot between low-cost sensor and governmental BAM–SMART calibration signal (green line)/BAM signal (blue line)/raw signal (red line).

**Figure 15 sensors-20-03617-f015:**
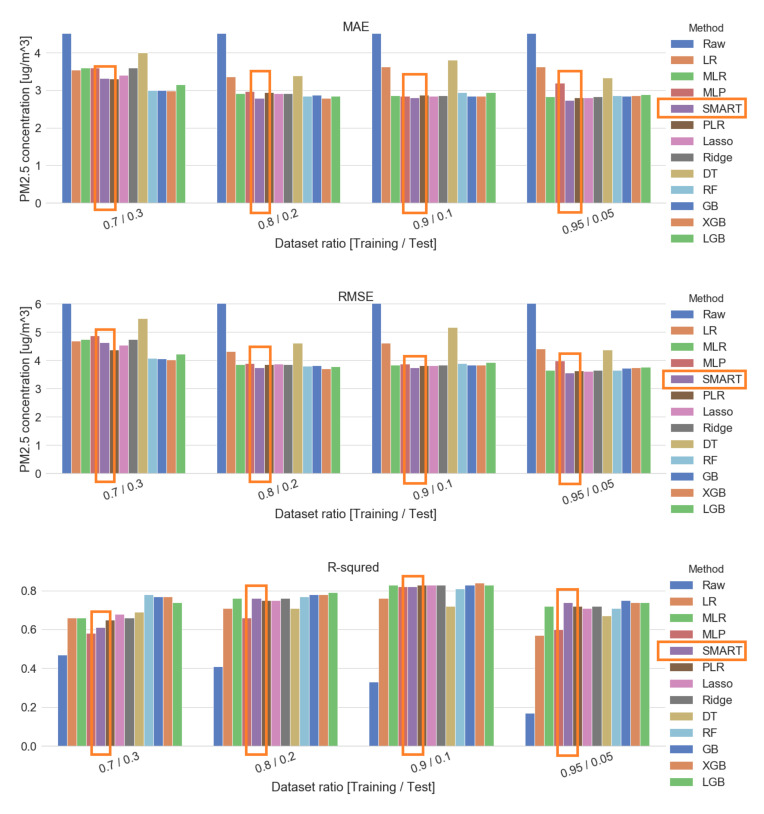
Comparison plot by metrics (sequential). GridsearchCV (10) found best hyperparameters as below. PLR (degree:2)/Lasso (alpha:5)/Ridge (alpha:100)/DT—decision tree (max depth = 12, min samples split =16)/RF (max depth = 6, min samples leaf = 8, min samples split = 24, n estimators = 500)/GB (learning rate = 0.05, n estimators = 200)/XGB (colsample bytree = 1, learning rate = 0.05, n estimators = 200, subsample = 0.3)/LGB (colsample bytree = 0.5, learning rate = 0.05, n estimators = 500, num leaves = 4, reg lambda = 10, subsample = 0.3).

**Table 1 sensors-20-03617-t001:** Chosen coefficients of multivariate linear regression (MLR) (80%—training dataset, 20%—test dataset, shuffled method, 5 min. sampling interval condition).

Raw(a)	Humidity	Temperature	Intercept
0.4470	−0.0581	0.0329	8.2511

**Table 2 sensors-20-03617-t002:** Hyperparameter of MLP (80%—training dataset, 20%—test dataset, shuffled method, 5 min. sampling interval condition).

Hidden Layer	Neurons/Layer	Epoch	Batch	Activation	Dropout Rate	Learning Rate	Optimizer
2	24	200	32	ReLU	0.2	0.005	Adam

**Table 3 sensors-20-03617-t003:** Metrics for performance analysis.

MAE	MSE	RMSE	R${}^{2}$
$\frac{1}{N}\sum_{i=1}^{N}\left|y_{i}-\hat{y_{i}}\right|$	$\frac{1}{N}\sum_{i=1}^{N}{\left(y_{i}-\hat{y_{i}}\right)}^{2}$	$\sqrt{\frac{1}{N}\sum_{i=1}^{N}{\left(y_{i}-\hat{y_{i}}\right)}^{2}}$	$1-\frac{\sum {\left(y_{i}-\hat{y_{i}}\right)}^{2}}{\sum {\left(\hat{y_{i}}-\bar{y}\right)}^{2}}$

$y:PM\underline{\;}reference,\;\hat{y}:PM\underline{\;}calibrated,\;\bar{y}:PM\underline{\;}mean\;of\;reference$.

**Table 4 sensors-20-03617-t004:** Comparison of calibration performance by input variables (short-term: 80%—training dataset, 20%—test dataset, shuffled method, 5 min. sampling interval condition).

Input Variables	Linear - ULR/MLR			Nonlinear - MLP		
	MAE	MSE	R${}^{2}$	MAE	MSE	R${}^{2}$
[uncalibrated] Raw PM	9.78	216.89	0.52	9.78	216.89	0.52
[ calibrated] Raw PM	3.69	24.44	0.78	3.55	23.12	0.80
[ calibrated] Raw PM + Humidity	3.11	18.72	0.84	2.99	16.69	0.84
[ calibrated] Raw PM + Temp	3.22	19.56	0.83	3.11	18.39	0.83
[ calibrated] Raw PM + Light	3.39	21.40	0.81	3.23	18.97	0.84
[ calibrated] Raw PM + Humidity + Temp	3.11	18.70	0.84	2.95	16.91	0.83
[ calibrated] Raw PM + Humidity + Light	3.09	18.61	0.84	2.99	17.01	0.83
[ calibrated] Raw PM + Temp + Light	3.19	19.25	0.83	3.10	18.15	0.83
[ calibrated] Raw PM + Humidity + Temp + Light	3.08	18.41	0.84	2.93	16.76	0.83

**Table 5 sensors-20-03617-t005:** Comparison of calibration performance by input variables (long-term: 80%—training dataset, 20%—test dataset, shuffled method, 5 min sampling interval condition)

Input Variables	Linear - ULR/MLR			Nonlinear - MLP		
	MAE	MSE	R${}^{2}$	MAE	MSE	R${}^{2}$
[uncalibrated] Raw PM	15.87	573.23	0.41	15.87	573.23	0.41
[ calibrated] Raw PM	4.28	33.79	0.82	4.21	33.79	0.79
[ calibrated] Raw PM + Humidity	4.01	30.13	0.84	4.04	32.15	0.77
[ calibrated] Raw PM + Humidity + Temp.	4.00	29.90	0.84	3.52	23.88	0.86

**Table 6 sensors-20-03617-t006:** Comparison of performance by sampling intervals (5 min/1 h/24 h: 80%—training dataset, 20%—test dataset, shuffled method).

Sampling Interval	Metric	Raw	LR	MLP	SMART
5 min	MAE	15.87	4.00	3.52	3.32
	MSE	573.23	29.90	23.88	22.06
	R${}^{2}$	0.41	0.84	0.86	0.89
1 h	MAE	14.72	3.68	3.29	3.51
	MSE	486.26	25.22	21.29	25.75
	R${}^{2}$	0.41	0.85	0.88	0.86
24 h	MAE	12.33	2.71	2.92	2.68
	MSE	299.55	21.72	29.62	21.99
	R${}^{2}$	0.37	0.77	0.75	0.77

**Table 7 sensors-20-03617-t007:** Metric analysis for data preprocessing methods (shuffled/sequential—5 min sampling interval condition).

Dataset Ratio	Metric	Shuffled					Sequential				
		PM Only		PM+Humidity+Temp			PM Only		PM+Humidity+Temp		
		Raw	LR	MLR	MLP	SMART	Raw	LR	MLR	MLP	SMART
70%/30%	MAE	15.68	4.25	3.98	3.65	3.29	8.92	3.54	3.60	3.60	3.32
	MSE	563.90	33.45	29.61	25.27	21.80	182.31	21.99	22.49	23.70	21.56
	R${}^{2}$	0.41	0.82	0.84	0.83	0.89	0.47	0.66	0.66	0.58	0.61
80%/20%	MAE	15.87	4.28	4.00	3.52	3.32	9.06	3.36	2.91	2.97	2.79
	MSE	573.23	33.79	29.90	23.88	22.06	196.35	18.70	14.84	15.20	14.02
	R${}^{2}$	0.41	0.82	0.84	0.86	0.89	0.41	0.71	0.76	0.66	0.76
90%/10%	MAE	15.8	4.34	4.06	3.47	3.23	11.67	3.62	2.86	2.84	2.80
	MSE	570.1	34.80	30.76	22.70	20.85	311.90	21.31	14.73	15.06	14.05
	R${}^{2}$	0.42	0.81	0.84	0.87	0.90	0.33	0.76	0.83	0.82	0.82
95%/5%	MAE	15.44	4.40	4.09	3.64	3.35	10.07	3.63	2.83	3.19	2.74
	MSE	549.64	36.53	31.96	24.63	22.48	194.54	19.44	13.34	15.92	12.75
	R${}^{2}$	0.42	0.80	0.83	0.86	0.88	0.18	0.57	0.72	0.60	0.74

**Table 8 sensors-20-03617-t008:** Metric analysis of various calibration methods (sequential method, 5 min sampling interval condition).

Data Set Ratio	Metric	Raw	LR	MLR	MLP	SMART	PLR	Lasso	Ridge	DT	RF	GB	XGB	LGB
70%/ 30%	MAE	8.92	3.54	3.60	3.60	3.32	3.31	3.40	3.60	4.00	3.00	3.00	2.98	3.15
	MSE	182.31	21.99	22.49	23.70	21.56	19.21	20.71	22.49	30.18	16.69	16.45	16.26	17.82
	R${}^{2}$	0.47	0.66	0.66	0.58	0.61	0.65	0.68	0.66	0.69	0.78	0.77	0.77	0.74
80%/ 20%	MAE	9.06	3.36	2.91	2.97	2.79	2.94	2.92	2.91	3.39	2.85	2.88	2.79	2.84
	MSE	196.35	18.70	14.84	15.20	14.02	14.80	14.98	14.84	21.24	14.43	14.58	13.80	14.26
	R${}^{2}$	0.41	0.71	0.76	0.66	0.76	0.75	0.75	0.76	0.71	0.77	0.78	0.78	0.79
90%/ 10%	MAE	11.67	3.62	2.86	2.84	2.80	2.87	2.85	2.86	3.81	2.95	2.85	2.85	2.95
	MSE	311.90	21.31	14.73	15.06	14.05	14.67	14.61	14.73	26.74	15.11	14.78	14.71	15.54
	R${}^{2}$	0.33	0.76	0.83	0.82	0.82	0.83	0.83	0.83	0.72	0.81	0.83	0.84	0.83
95%/ 5%	MAE	10.07	3.63	2.83	3.19	2.74	2.81	2.80	2.83	3.33	2.86	2.84	2.86	2.89
	MSE	194.54	19.44	13.34	15.92	12.75	13.21	13.01	13.34	19.20	13.37	13.88	14.07	14.14
	R${}^{2}$	0.17	0.57	0.72	0.60	0.74	0.72	0.71	0.72	0.67	0.71	0.75	0.74	0.74

**Table 9 sensors-20-03617-t009:** Performance comparison by group & method.

Category	Metric	Other Group – Shuffled (MLR)	Our Group – Shuffled (SMART)	Our Group – Sequential (SMART)
Before calibration	MAE	5.8	15.1	11.4
	RMSE	7.5	23.1	17.3
After calibration	MAE	3.2	3.4	2.8
	RMSE	4.1	4.8	3.7
	R${}^{2}$	0.57	0.89	0.81

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
