# Peer review of "Long-Term Evaluation and Calibration of Low-Cost Particulate Matter (PM) Sensor"

_sensors, 2020, doi:10.3390/s20133617_

Round 1
Reviewer 1 Report
In this paper, the authors co-located Plantower PMS7003 low-cost optical particle counter with a regulatory BAM monitor, and compared the performances of multiple calibration methods including a hybrid SMART method. Overall, this study is comprehensive and contains lots of information that are useful for other researchers working on low-cost sensors. There are a few places that may be revised to help readers to better understand the contents.
Most comparisons were performed using 5-min average readings from the BAM monitor. High-resolution BAM measurement data are noisy. That’s why usually 1-hour average data are used even though higher-resolution data are available. It would be helpful to clearly acknowledge this limitation.
The developed SMART calibration method is innovative, its performance, statistically, is slightly better than MLR and MLP, but not substantially better. In addition, it seems that the SMART method may apply different underlying method to calibrate sensor data collected within one day, which may lead to other unwanted issues such as discontinuation of error structure. It would be helpful to clearly acknowledge the limitations of the SMART methods.
In Figure 4, it seems that a standard PM10 inlet was attached to BAM with no size selective inlet such as sharp cut cyclone. Could the authors please confirm that the BAM actually measures PM2.5 but not PM10? or perhaps size separation was performed inside the instrument enclosure?
In Table 1, the coefficient for “PM25 a raw” is 0.4470, does this suggest PMS7003 under-estimate PM2.5 concentrations by more than a factor of 2?
Some tables and figures may not be necessary. For example, figures 7, 8 and 12 are explanations of a few well known statistical phenomenon. They are helpful but they do not seem to be important.
In Figure 10, the maximum temperature shown is 74, I assume the unit is not Celsius. But if the unit is Fahrenheit, it would seems that some data points are missing because the highest temperature in summer time would be much higher than 74 Fahrenheit.
In lines 252-253, the author mentioned that “high R2 among the low-cost sensors means a calibration model can be shared in low-cost sensors together”. This is not true. High R2 among sensors are expected, and it only suggest good sensor precision, not accuracy. If the sensor were to be placed at different locations, its accuracy may change due to changes in particle compositions etc.
The author also mentioned a few times in the article (line 3 and line 24) that seems indicating BAM is the only regulatory PM measurement instrument. It would be helpful to revise the wording to avoid confusions.
Author Response
I appreciate your comments. The comments are very valuable and I appreciate your interest in my paper. Your deep insight into the review is very helpful with revision and widens my horizons. Your comments were very reasonable and I tried to properly reply to your comments.
"Please see the attachment."

Reviewer 2 Report
This research study is focused on the evaluation of a new calibration (SMART) proposed for low cost sensor of PM. Results achieved by these instruments are compared with the ones obtained from BAM operating in parallel and in the same condition. The performance of this calibration is compared the efficiency of other calibrations. Authors underlined the efficiency of this calibration in increasing the reliability of these low cost sensors.
In general i think that the research study is focused on a very crucial aspect, but the manuscript needs more attention. An intensive check of english grammar and style is strongly suggested in the text. This will increase the comprehension of the results and their discussion. Furthermore i think that results obtained in this study need more explanation. Finally i kindly suggest a general re-organization of the text(since in some point it is difficult to follow up the discussion).

Author Response
I apologize for some confusing sentences and unclear explanations, and deeply appreciate your close review and comment including English grammar and style on the contents.
We have rechecked the entire paper and re-organized the paper, especially the introduction. Also, some contents were moved into an appendix in order to build a more balanced structure. We hope that this revision satisfies your expectations.
"Please see the attachment."

Reviewer 3 Report
This paper has a rather catchy title and the introduction really attracted my reading. However, the first half of the document is very difficult to follow. It seems that the paper has been written by multiple authors who did not read each other parts. As a complement to my attach comments, I do advise the authors to say from the beginning that the study has been carried out on PM2.5. However, the subject is very interesting and the approach seems rather novative. I have no doubt that this paper can be of interest after a very thorough and serious proofread. Maybe the authors can also partially reorder the paper as they often mentioned following paragraph to justify some choices. Congratulations to the authors for the idea.

Author Response
I deeply appreciate your valuable opinions on the paper and it can help us improve the maturity level of the paper. We thank the comments so that we can express terms more appropriately as well as English expressions and grammar.
I hope if I had done well with the opinions you pointed out. I wrote the revised paper with a revised line number corresponding to your comments.
"Please see the attachment."

Reviewer 4 Report
This paper provides an interesting contribution into calibration techniques for low-cost PM sensors with valuable results. The paper is well written and easy to understand to a broad audience. I believe the following minor changes will help improve the quality of the paper prior to final publication.
Please, could you add characteristics of the humidity and temperature sensor and the light sensor as they are used for the calibration. In particular, their resolution and precision?
2.1.2. Governmental BAM – High-end PM monitoring station
What are the characteristics of the reference instrument at 5 min resolution.
2.3. Data Calibration
First paragraph: Can you clarify more what are the two selection of parameters for? Ambient light does not appear in the rest of this subchapter.
Line 170 typo “approximation”
Line 173: the content of Figure 9 needs to be explained in the text and caption as the meaning of the different arrows are not self-explanatory.
Line 174: please explain in the text that the two axis used to generated the independent variable space are humidity and temperature.
Do the temperature and humidity sensors have a resolution that is good enough for the grid defined by the prevailing model map? What is the resolution of the grid used to generate the map?
Please provide more explanations about Figure 11 and add axis labels and units to the graphs on the right of the Figure.
Line 203: the k-kohler theory is evoked here but is it used afterwards? It is not clear why it is introduced here and how does it relates with the rest of the analysis conducted. Please link it more with the rest of the analysis.
For each analysis conduted (Line 209, line 222), please specify the time period used (5min/ 1 hour or 24h).
257 3.3. Comparative analysis [the low-cost sensor and governmental BAM] : after calibration
There were three sensors in the previous subchapter (before calibration) but here results are only presented for one sensor. Or are the results averaged over the three previous sensors? Please specify.
Line 269: any suggestion why the sequential methods obtained lower scores than the shuffled methods?
Table 11 and Figure 15 present the same results, I would suggest moving table 11 to the appendix.
295 3.5. Comparative analysis [similar previous study]
Could you include other similar studies to the comparison? For example (but not limited to):
- Malings, C. et al. Fine particle mass monitoring with low-cost sensors: Corrections and long-term performance evaluation. Aerosol Sci. Technol. 0, 1–15 (2019).
- Zusman, M. et al. Calibration of low-cost particulate matter sensors: Model development for a multi-city epidemiological study. Environ. Int. 134, 105329 (2020).
Hybrid Calibration is introduced only at the end of the conclusion. How is it different from rendez-vous calibration developed by previous studies? Arfire, A., Marjovi, A. & Martinoli, A. Model-based rendezvous calibration of mobile sensor networks for monitoring air quality. in 2015 IEEE SENSORS - Proceedings (2015). doi:10.1109/ICSENS.2015.7370258
Line 331 “This method is expected to maximize the accuracy of low-cost sensors up to the accuracy
of BAM.“
Could you compare the metrics obtained to the metrics used to certify reference grade instruments in order to further support his claim?
The figures would also benefit from the following changes, and maybe removal of some of the figures as some of them just present previously published work
Figures:
Figure 1: add a scale + what do the number means? Are they concentrations? Are the air quality index values?
Figure 2: very valuable to understand the content of the paper.
Figure 5: may be removed
Figure 7 and 8 may be removed
Figure 9: why is BAM barred in the lower half of the diagram?
Figure 13: please specify clearly which reading is from the BAM. And add units.
Table 8 and 9 and 10 could go in the appendix.
Check units throughout the figures and tables as well as the axis labels.
Overall this is a very worthwhile paper and I look forward to being able to cite it in future work.
Author Response
I appreciate your comments and interests. The comments are very valuable to revise the paper and I appreciate your strong interest in our paper. Your comments were very reasonable and I tried to properly answer all your comments. I hope it meets your expectation in our paper.
"Please see the attachment."

Round 2
Reviewer 2 Report
I think the authors increased the quality of the paper, thus modifying and improving the text and the presentation of this research study and its results.
On my side, the comments and the recommendations previously indicated in my first revision were satisfied. However i kindly suggest a moderate check on English language and style, in order to increase the comprehension of the text.
Reviewer 3 Report
The authors did improve the quality of the manuscript. They answered and considered the comments of my previous revision. I still suggest a moderate english check in order to remove the last language and style errors. Congratulations for your work.